# Quantification of Soil Organic Carbon in Biochar-Amended Soil Using Ground Penetrating Radar (GPR)

**Xiaoqing Shen** [1], **Tyler Foster** [1], **Heather Baldi** [1], **Iliyana Dobreva** [1], **Byron Burson** [2], **Dirk Hays** [1], **Rodante Tabien** [3] **and Russell Jessup** [1,*]

[1] Department of Soil and Crop Science, Texas A&M University, College Station, TX 77843-2474, USA; shenxiaoqing@tamu.edu (X.S.); tyler.foster@tamu.edu (T.F.); hdbaldi@tamu.edu (H.B.); iliyanad@tamu.edu (I.D.); dbhays@tamu.edu (D.H.)

[2] USDA-ARS, Southern Plains Agricultural Research Center, College Station, TX 77845, USA; Byron.Burson@ARS.USDA.GOV

[3] Texas A&M AgriLife Research Center, Beaumont, TX 77713, USA; retabien@ag.tamu.edu

\* Correspondence: rjessup@tamu.edu

**Abstract:** The application of biochar amendments to soil has been proposed as a strategy for mitigating global carbon (C) emissions and soil organic carbon (SOC) loss. Biochar can provide additional agronomic benefits to cropping systems, including improved crop yield, soil water holding capacity, seed germination, cation exchange capacity (CEC), and soil pH. To maximize the beneficial effects of biochar amendments towards the inventory, increase, and management of SOC pools, nondestructive analytical methods such as ground penetrating radar (GPR) are needed to identify and quantify belowground C. The use of GPR has been well characterized across geological, archaeological, engineering, and military applications. While GPR has been predominantly utilized to detect relatively large objects such as rocks, tree roots, land mines, and peat soils, the objective of this study was to quantify comparatively smaller, particulate sources of SOC. This research used three materials as C sources: biochar, graphite, and activated C. The C sources were mixed with sand—12 treatments in total—and scanned under three moisture levels: 0%, 10%, and 20% to simulate different soil conditions. GPR attribute analyses and Naïve Bayes predictive models were utilized in lieu of visualization methods because of the minute size of the C particles. Significant correlations between GPR attributes and both C content and moisture levels were detected. The accuracy of two predictive models using a Naïve Bayes classifier for C content was trivial but the accuracy for C structure was 56%. The analyses confirmed the ability of GPR to identify differences in both C content and C structure. Beneficial future applications could focus on applying GPR across more diverse soil conditions.

**Keywords:** ground penetrating radar; biochar; attribute analysis; machine learning

## 1. Introduction

The transition of global land resources into managed agricultural systems has greatly changed the terrestrial C balance, and this phenomenon has been accelerated in recent decades by the increase in human population and demands for food, feed, fiber, and fuel [1,2]. Land use changes into cropland and grassland systems have resulted in a significant loss of soil organic carbon (SOC), which is the dominant component of soil organic matter (SOM). An estimated 30 to 60 Pg (Pg = $1 \times 10^{-12}$ kg = 1 billion metric tons) of SOC has been lost over the past 100 years [3,4], and cumulative historic SOC losses of up to 230 Pg also have been reported [4,5].

Due to the importance of SOC in the carbon (C) cycle and its specific capacity for C dioxide ($CO_2$) sequestration, novel strategies for increasing the SOC pool are critical. Incorporation of stable, recalcitrant C into soils via biochar amendments is one potential approach. Biochar, made by pyrolysis, a thermal decomposition process of burning organic materials at high temperatures in the absence of oxygen, has varied physicochemical qualities that provide differentiated effects upon the soil environment when applied [6]. Being highly recalcitrant to decomposition, biochar can significantly decrease the rate at which photosynthetically fixed C is returned to the atmosphere [7,8]. In addition to $CO_2$, biochar also can offset other greenhouse gases including nitrous oxide and methane [9–11].

To maximize the beneficial effects of biochar amendment applications towards global SOC pools, a rapid, nondestructive, and inexpensive method to detect and quantify belowground C is needed. Conventional means used to quantify SOC involve coring and probing for soil samples, followed by diverse chemical assays to analyze those samples. Significant soil disturbances occur during the collection and handling of the samples, and oxidation, volatilization, microbial degradation, and other sampling biases also often occur [12–14]. Ground penetrating radar (GPR) can serve as an alternative to traditional extraction methods by providing a convenient, nondestructive, and relatively inexpensive means of determining SOC. Much of the previous GPR research has focused on material identification spanning civil engineering and archaeology [15,16]. These GPR applications have focused primarily on relatively large objects such as items at archaeological sites, land mines, rocks, tree roots, and groundwater [17–22]; however, detecting organic C in the soil profile using GPR will be challenging. Other research also has shown that soil water content impacts dielectric properties and increasing water content results in both decreasing radar velocity and increasing attenuation [23,24].

Despite these challenges, researchers have successfully used GPR to estimate C stocks in wetlands [14]. Zheng et al. [24] proposed that SOC density can be estimated using GPR. Peat soils also have been evaluated with GPR to determine the C morphology, volume, and thickness [25,26]. These examples provide a rationale and framework to develop novel GPR methods for biochar and SOC quantification across major soil types worldwide [27–29]. No previous research has focused on estimating soil organic carbon in a biochar form using GPR. When utilizing artificial soil media consisting of silicon sand and different C sources as proxies for native soils, the signal return will be limited to those C sources. Such an experimental environment would provide an opportunity to adapt identification methodologies to analyze collected data. However, the SOC particle size is considerably smaller than most experimental objects targeted by GPR in civil engineering and archaeological studies. Visualization of such minute objects is debatable; therefore, a quantification tool such as attribute analysis is needed.

Attribute analysis was first applied by the seismic industry. It extracts features from the signals received and the signal provides a better interpretation of 2-D and 3-D GPR data [18]. An entire GPR data set is termed a B-scan, and one column of the data is known as an A-scan. In some cases, an A-scan is referred to as a trace. Normally, analyses conducted based on A-scans are referred to as attribute analysis. Some attributes, such as instantaneous amplitude, phase, and frequency, have been used in seismology [18]. Moreover, when applying attribute analysis on GPR data, more attributes such as relative reflectivity, phase relationships, complex trace attributes, and amplitude variation with offset (AVO) are introduced to fit the character of the GPR data [18,30]. Attribute analysis is a potential quantification method for targeting a subtle object's, such as organic C, GPR frequency change. For this study, maximum amplitude, intensity, energy, and area of the GPR data on the A-scan base were chosen to conduct attribute analysis similar to previous reports [20,30,31].

The Naïve Bayes classifier applies Bayes' theorem with the assumption that all features are strongly independent [32,33]. It is a simple probabilistic classifier applied to predictive modeling or machine learning [34,35]. With assigning training and validation data sets, a predictive model could predict C content and structure using GPR data as a supervised machine learning model. In this study, our objectives were to (1) detect underground biochar amended soil C content utilizing GPR, (2) correlate

soil moisture with GPR data, and (3) develop novel GPR analysis tools to assess SOC quantification with the use of attribute analysis and the Naïve Bayes predictive model.

GPR visualization methods such as wavelet analysis, in contrast, are based on signal data collected by variety of sensors. For GPR it is effective at filtering noise and analyzing signals [36,37]. With wavelet transformation of signals, their time and frequency domain change corresponding to the scale chosen. With control of the scales, different scales of wavelets will pass through the data collected while generating a coefficient [36]. There are two types of wavelet transforms, continuous wavelet transform and discrete wavelet transform. Continuous wavelet transform allows the translating and scale parameter of wavelets to vary continuously, and discrete wavelet transform allows wavelets sample discretely [38]. Application of wavelet analysis to GPR data and plotting time–frequency domain allows the signal change to be visualized. By changing wavelet scale, different time–frequency plots can be generated [36]. With visualization of the signal change, and the coefficient generated by different scale, different carbon percentages could possibly be visualized.

## 2. Materials and Methods

### 2.1. Material Preparation and Treatments

The biochar used was produced from pearl millet-Napier grass (*Pennisetum glaucum* [L.] R. Br. x *Pennisetum purpureum* Schumach) culms and leaves pyrolized at 400 °C. Other C sources used were coconut (*Cocos nucifera* L.) shell granular activated C (psc 1240, Prominent Systems Inc.) and graphite. In this study, each C-sand mixture represented a treatment for a total of 12 treatments. A mixture of 50% activated C and 50% graphite was made by mixing 1:1 with sand, and this was the same approach used to make the 100% activated C and 100% graphite. Other treatments were made by mixing sand and biochar at eight different ratios: 0%, 2%, 4%, 6%, 8%, 10%, 50%, and 100%. Table 1 lists the summary of the treatments and the C content. The 0% biochar contained pure sand. The C content of each source material was 54.2% for biochar, 80% for activated C, 100% for graphite, and 0% for sand. Treatments with less than 10% C–sand mixture were included in the study because they represent the approximate average belowground organic C percentages. Furthermore, 50% and 100% were chosen to compare the GPR performance on aggregated C with different C structures. Different percentage mixtures were made by mixing the corresponding amounts of materials with pure sand by weight and subsequently filling the mixture into sample containers. Three replications of each treatment were made where each sample was placed into a container composed of a silicone sandwich bag (Stasher, Inc.; Emeryville, CA, USA) 19.05 cm x 17.78 cm x 2.54 cm in size. A hole 1 cm in diameter was punched into the bottom left corner of each bag to allow a 6 cm soft silicone tube to be attached and a silicone ring was used to secure the tube. This modification provided a means for adding water into the bag without opening it multiple times. Three moisture levels, 0 %, 10 %, and 20 %, were evaluated. The water in each moisture level was added to the sample containers 24 h before the data collection. For each moisture level, the antenna scanned the trough six times.

**Table 1.** Summary for 12 treatments and the carbon content both by weight and percentage for each treatment.

| Treatment Number | Treatment | Carbon Content by Weight Per Sample (g) | Carbon Content by Percent Per Sample (%) |
|---|---|---|---|
| 1 | 0% Biochar | 0 | 0 |
| 2 | 2% Biochar | 7.5012 | 1.0840 |
| 3 | 4% Biochar | 13.6150 | 2.1680 |
| 4 | 6% Biochar | 18.7640 | 3.2520 |
| 5 | 8% Biochar | 24.0214 | 4.3360 |
| 6 | 10% Biochar | 28.9428 | 5.4200 |
| 7 | 50% Biochar | 67.7500 | 27.1000 |
| 8 | 100% Biochar | 86.7200 | 54.2000 |
| 9 | 50% Graphite | 250 | 50 |
| 10 | 100% Graphite | 430 | 100 |
| 11 | 50% Activated Carbon | 200 | 40 |
| 12 | 100% Activated Carbon | 256 | 80 |

*2.2. Data Collection*

Data collection was conducted at the Texas A&M Agricultural Research Farm in Snook, Texas (30° 35' 49.74'' N, 96° 20' 52.44'' W) on a 20 m x 2.75 m x 1.5 m aboveground trough filled with dry, pure quartz sand (Figure 1). Three replications were used in a randomized complete block design. Every treatment was buried under 100% sand at a depth of 5.08 cm and spaced 25.4 cm apart along a single, medial line in the trough. Due to repeated radar cart traffic across the trough during the experiment, some sample bags' burial depth varied slightly. In addition, retaining the exact same start point across successive scans was difficult and required only a central subset of three A-scans within each sample to be analyzed.

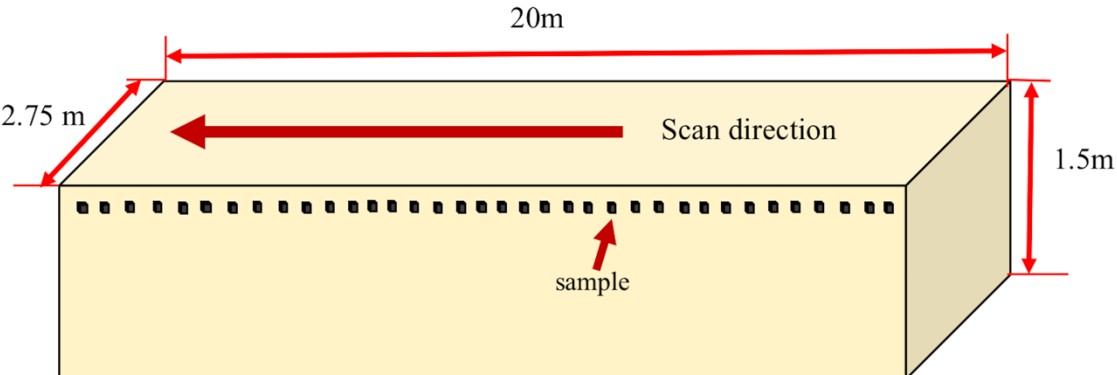

**Figure 1.** The experiment sand trough illustration diagram. All the treatments (samples) were buried 5.08 cm in depth and 25.4 cm apart. The dimensions of each sample are 19.05 cm x 17.78 cm x 2.54 cm.

The GPR antenna system utilized in this study was developed by IDS GeoRadar. The system and antenna alignment were developed based on previous preliminary research results [21]. The system is a down-looking 7-channel multiarray. The channel configuration consists of four transmitting and four receiving antenna separated at 4 cm intervals as seen in Figure 2. This spacing fulfills the Nyquist criterion for data acquisition, utilizing this sensor with the central frequency of 1.8 GHz. This dense array allows for imaging underground objects and potentially identifying subtle features such as soil C. The antenna was assembled to a scanning cart (Figure 3). Each channel generated one B-scan with 1 cm x 1 cm pixel resolution following a transect of the trough. Each scan gave seven B-scans of all 36 plots, out of three replications for twelve treatments. A processed B-scan and a raw B-scan in heat-map form was used to help visualize the collected data (Figure 1).

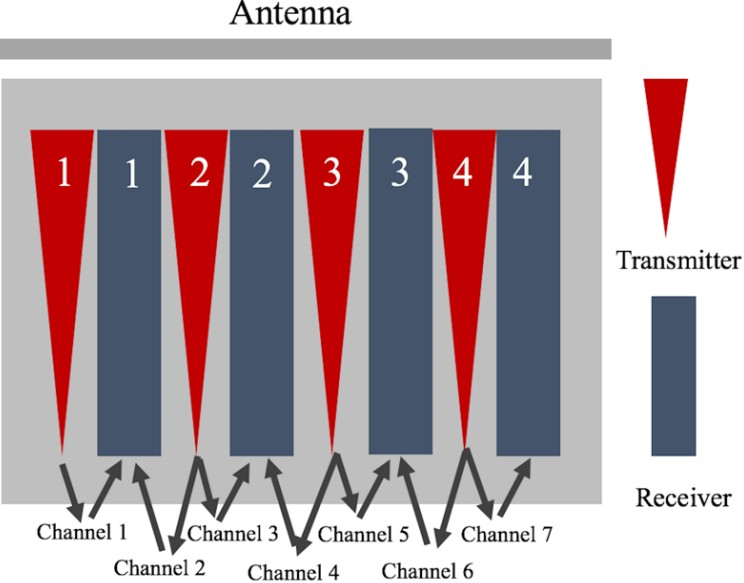

**Figure 2.** The display of seven channels created by transmitters and receivers used to collect ground penetrating radar (GPR) data. The orange triangles represent transmitters and the dark blue rectangles represent receivers. The antenna used contains seven channels with each channel consisting of one transmitter and one receiver.

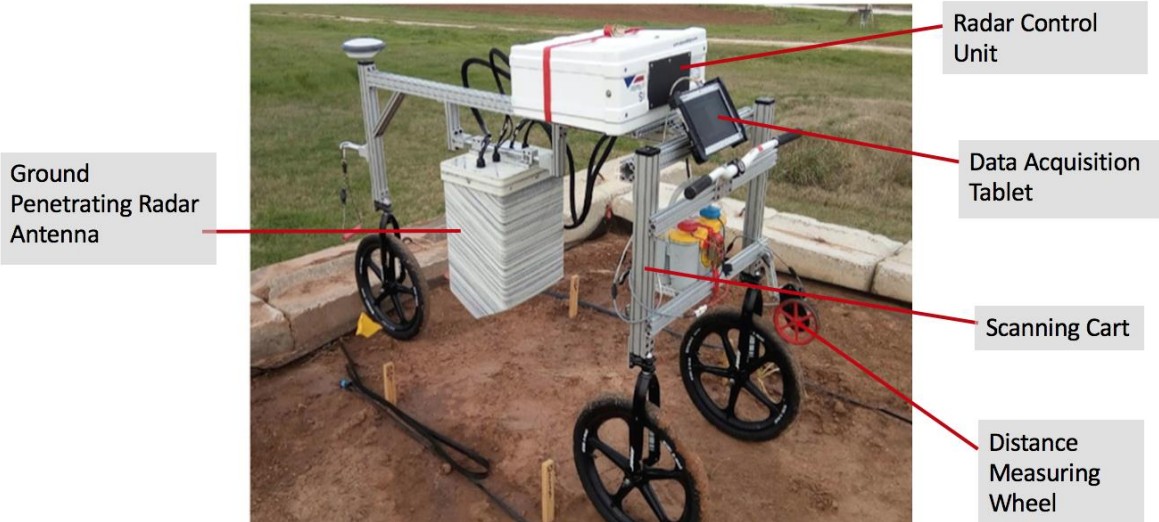

**Figure 3.** The scanning cart prototype equipped with Ground Penetrating Radar (GPR).

### 2.3. Data Processing and Analyses

No standard procedure has been reported or developed to preprocess GPR data for non-visualized quantification of small targets such as organic C. Because of this, preprocessing procedures were reduced to a minimum, including automatic surface removal and fast Fourier transform bandpass filtering. This was performed in GPRLib, which is a custom Python library still under development. This process removed the noise of the data above the sand surface and was followed by sub-setting the bottom of the B-scan by adding 50 rows from the sand surface. This was required due to small variations in sample bags burial depth in the trough. Sub-setting the data with a wider range of rows ensured inclusion of all of the samples. Bandpass function was conducted after surface removal, while the Fourier transfer bandpass filter set the lower band frequency to 1.0 GHz and higher band frequency to 3.2 GHz to block signals out of this frequency range. After preprocessing, attribute analysis was

performed using a novel Python script where maximum amplitude, intensity, energy, and area were calculated. All attributes were transformations of the signals' amplitudes on the central three A-scans of each sample based on location. The A-scans were selected for the middle three columns of each sample, which possibly represent the most precise location of the samples. After computing the attributes on the three columns, we averaged the three and got the final attribute value. The reason is that the location of each sample is very hard to tell on the B-scan, so the location of samples was estimated based on where the samples were buried. To avoid the error of picking wrong A-scans, the central three A-scans were computed and averaged. Maximum amplitude was computed by finding the maximum value of the A-scan amplitude. Intensity was computed by finding the maximum value of the squared amplitude on the A-scan, while area was computed by finding the maximum value of the integrated amplitude on the A-scan. Lastly, the equation (1) denoted how to calculate the energy. Energy was computed by determining the maximum value of the integrated squared amplitude as suggested in [30].

$$\int_0^{t_{end}} A^2 dt \tag{1}$$

After completing this workflow, the data set was exported to a spreadsheet for statistical analyses and predictive modeling. Continuous wavelets analysis was conducted on the data, with 150 scales from 1 to 150 passed to PyWavelets package on Python v. 3.6.5 (Python Software Foundation, 2018). Wavelet coefficients were generated for the 150 scales and 12 treatments as well as the p-value.

*2.4. Statistical Analysis*

A total of 3868 data points were collected, although the total number should be 4536 calculated by 12 treatments, three replications, seven channels, three moisture levels, and six times scan per each moisture level. There were 668 data points missing because a few scans failed to collect data through the end of trough. The mean differences among the seven channels, moisture levels, and treatments were analyzed based on the attributes. The four attributes were the four variables, while the C content in percentages and moisture levels were the responses. A Naïve Bayes predictive model also was developed and validated based on the entire data set. To determine the performance of the seven channels, analysis of variance (ANOVA) and interquartile range (IQR) were conducted. Spearman's rank correlation tests and Pearson correlation tests were conducted to explain the moisture level differences and treatment differences. For moisture levels, Spearman's rank correlation analyses were conducted to assess the relationship between two variables using a monotonic function since moisture level was a categorical variable and the attributes were continuous variables. As for treatment differences, the three C sources were separated and recombined together, and a Pearson correlation test was conducted to determine the linear relationship between C content and attributes.

To further assess the ability of GPR to identify different C contents and C structures, two Naïve Bayes predictive models were constructed on data collected by channel 4 to prevent data overlapping between channels during machine learning. The Naïve Bayes model uses Bayes' theorem with an independence assumption between features to construct a classifier. If a large amount of data for different percentages of organic C is collected, the predictive model can develop a baseline for those C percentages and can predict C content of an in situ field. The same procedure also can be applied to differentiate C structures. To classify the different C contents, classification of the model was: 0%, 2%, 4%, 6%, 8%, 10%, 50%, and 100% biochar, 50% and 100% activated C, and 50% and 100% graphite. To classify the different carbon structure, the data was divided by two classes: Biochar and Non-Biochar. The parameter used to construct the model was energy because the assumption of a Naïve Bayes classifier requires non-dependent variables. Since all the attribute variables were dependent with each other, only one attribute was put into the model. To validate the accuracy of the model, the data was separated into a 75:25 ratio as a training and validation set. This ratio was used because the data set sample size was not large enough to support a 50:50 ratio for the 12 classifiers. The same model procedure was iterated 1000 times for each model, and the data was split differently for every iteration.

The confusion matrix and the accuracy computed from the confusion matrix were used to estimate the ability of the model to distinguish different classes. Usually accuracy under 50% (random guess) was considered trivial.

All statistical analyses were completed on RStudio v.3.5.2 (RStudio Team, Boston, MA, 2019).

## 3. Results

Data from channels 1, 6, and 7 were discarded because of the large number of extreme outliers when compared to the other channels. The extreme outliers were identified as exceeding three times the IQR. This indicates the border transmitters 1 and 4 (Figure 2) were affected more by their locations. This resulted in more noise being collected by these channels which produced highly variable data. The data set after excluding channels 1, 6, and 7 consisted of 2208 data points.

### 3.1. Moisture Levels

After confirming significant mean differences of attributes for each moisture level, the variance in the attributes was used to establish a correlation between moisture level and the four attributes separately since moisture level was a categorical variable and the attributes were continuous variables. Spearman's rank correlation analyses were conducted, and they assessed the relationship between two variables using a monotonic function. With a positive correlation coefficient, the attribute increased when the moisture level increased. Correlations were considered significant at $p \leq 0.05$, 0.01, and 0.001. The correlation coefficient results (Table 2) showed the coefficients of the association between moisture levels and the attributes, as well as the $p$-values. All coefficients were positive but generally low with significant $p$-values. The correlation coefficients were low because the data was variable since the sub-grouping data was from 12 treatments for each moisture level. Therefore, for all attributes, they increased when the moisture level increased, which indicates that GPR data had a positive relationship with moisture level. This can be explained by C aggregation. Because water and C aggregate, the target object became larger at a higher moisture level, which made it easier to detect C with GPR. Coefficient between energy and moisture levels was the highest, indicating that energy better explained the correlation between moisture level and GPR data.

**Table 2.** Spearman's rank correlation coefficient between attributes and moisture levels at 0%, 10%, and 20% in 12 carbon–sand mixture samples. All attributes were calculated based on the amplitude of the received GPR signals.

| Attributes | Spearman's Correlation Coefficient | *p*-Value |
|---|---|---|
| Maximum amplitude | 0.1930 | $<2.20 \times 10^{-16}$ *** |
| Intensity | 0.1627 | $1.456 \times 10^{-14}$ *** |
| Area | 0.0874 | $3.912 \times 10^{-5}$ *** |
| Energy | 0.2278 | $<2.20 \times 10^{-16}$ *** |

*Note:* *, **, and *** denote significance at 0.05, 0.01, and 0.001 level of probability, respectively.

### 3.2. Treatment Differences

The Pearson correlation coefficients between graphite C content and attributes indicated all attributes had significant linear correlations with graphite with all $p$-values less than 0.001 (Table 3). The correlation coefficients between the three attributes (maximum amplitude, intensity, and area) and graphite C content were all positive, indicating the value of these three attributes increases when graphite C content increases. Regarding the attribute energy, the correlation coefficient with graphite C content was negative ($-0.4138$); therefore, energy decreases when activated C content increases, and it was the strongest correlation for C content. The negative relationship could be caused by the method used to calculate energy. The energy was estimated from a squared signal amplitude. Therefore, the attribute energy contained more information than the other three attributes.

**Table 3.** Pearson correlation between attributes and graphite carbon content. All attributes were calculated based on the amplitude of the received GPR signals.

| Attributes | Pearson Correlation Coefficient | *p*-Value |
|---|---|---|
| Maximum amplitude | 0.2822 | $3.706 \times 10^{-12}$ *** |
| Intensity | 0.2821 | $3.803 \times 10^{-12}$ *** |
| Area | 0.3501 | $<2.20 \times 10^{-16}$ *** |
| Energy | −0.4138 | $<2.20 \times 10^{-16}$ *** |

*Note:* *, **, and *** denote significance at 0.05, 0.01, and 0.001 level of probability, respectively.

The criteria for grouping biochar and activated C was based on the significance of the paired Tukey's honestly significant difference (HSD) test between each treatment (data not shown). For each attribute, the biochar treatment was selected when it was significantly different from 50% and 100% activated C. Some biochar treatment attributes were not significantly different from another treatment. This could be due to the C content of the biochar used in a given treatment being low. For instance, the C content for 2% biochar was 1.08% and GPR was not able to detect the signal change. The attributes in such cases could be reflecting the signals from the silicone sample bag or the sand. The Pearson correlation coefficients between the sub-groups (activated C and biochar C content) and the four attributes revealed that all attributes had a significant linear correlation with the *p*-value less than 0.001 (Table 4). The correlation coefficients between attributes, except for the attribute energy and graphite C content, were negative. This shows that those attributes decreased when the graphite C content increased. As for energy, the correlation coefficient with graphite C content was 0.1678 and indicated energy increased when activated C content increased. Note the correlation coefficients strengthened the negative correlation for the three attributes but reversed the relationship for energy (Table 4).

**Table 4.** Pearson correlation between attributes and carbon content of biochar-activated carbon subgroup. All attributes were calculated based on the amplitude of the received GPR signals.

| Attributes | Pearson Correlation Coefficient | *p*-Value |
|---|---|---|
| Maximum amplitude | −0.2316 | $6.911 \times 10^{-15}$ *** |
| Intensity | −0.2114 | $1.338 \times 10^{-12}$ *** |
| Area | −0.2408 | $5.252 \times 10^{-16}$ *** |
| Energy | 0.1678 | $1.378 \times 10^{-7}$ *** |

*Note:* *, **, and *** denote significance at 0.05, 0.01, and 0.001 level of probability, respectively.

*3.3. Naïve Bayes Predictive Model*

The predictive accuracy for C content was insignificant. One possible reason for this finding is that the limited data points for 12 classes, and some classes like 100% biochar and 50% graphite, have very similar C content and are misclassified by the model.

The predictive accuracy for C structure was 56.12% averaged after 1000 iterations. The confusion matrix for a single random iteration was displayed to illustrate how to compute the accuracy (Table 5). The accuracy was calculated by adding the corrected prediction and divided by the total number, which was 60.29%. Even though 56% was still close to 50%, the accuracy for C structure was more promising than C content since there were two classes compared to 12. This indicates that within biochar and non-biochar, while the C content is different, the attribute energy is still capable of identifying the two.

**Table 5.** The confusion matrix after one iteration for predicting carbon structure using Naïve Bayes classifier.

|  |  | Reference | |
| --- | --- | --- | --- |
|  |  | Biochar | Non-Biochar |
| **Prediction** | Biochar | 64 | 42 |
|  | Non-Biochar | 12 | 18 |

*3.4. Wavelet Analysis*

No significant wavelet coefficients were found on the entire data set, indicating that carbon was too small for GPR to visualize.

**4. Discussion**

The primary purpose of this study was to explore the feasibility of using GPR to detect fine particles such as SOC. To test this assumption, three different C sources (biochar, activated C, and pure C structure graphite) were mixed at different percentages with sand and characterized with GPR. Our results indicated such feasibility. The GPR was able to detect biochar-amended soil and the predictive model differentiated C structures with approximately 56% accuracy. In somewhat similar studies [25,26], peat land soil thickness and clay content were evaluated and it was determined that GPR can be used under different soil conditions. Wavelet analyses were uninformative, suggesting that the method is likely not an effective strategy to quantify SOC.

The detection of soil moisture variation in our study further indicates the utility of deploying GPR in more complex soil conditions. The significant correlations between moisture level and attributes would be important for other research aiming to characterize soil conditions using GPR. Previous researches have already begun to develop tools towards achieving this goal [39,40]. This indicated that if we are to further study soil moisture, multidimensional attribute analysis might be more suitable to display spatial distribution than attribute analysis.

Our findings also provide a framework for predicting and quantifying soil C content. By successfully differentiating soil amendments in sand, these findings could facilitate further research to combine remote sensing tools, like GPR, with agriculture to better understand and utilize belowground resources. This research can serve as a framework for future efforts to predict and quantify C.

Pearson correlation significantly correlated attributes with C content, but the highest correlation was a negative correlation between graphite and energy. The negative correlation might be the result of error during data collection or the fact that subgroup treatments based on Tukey's HSD test would bring the most potentially different combinations which resulted in the most significant and precise correlation. Moreover, since the attribute energy was transformed the most from the original data, it might possess some unwanted data which could affecting the results. But for biochar and activated C, energy was the only attribute that had a significant Tukey's HSD test (data not shown), indicating energy could be a useful predictor as long as redundant data is removed. Among all the correlation coefficients, the coefficient between graphite C content and energy was the highest. This indicates the energy of GPR data responded stronger to graphite than the other C sources.

Because the C content would not be as high as graphite in either in situ soil conditions or biochar-amended soils, the Pearson correlations might not be effective indicators when C content is low. Utilizing machine learning models like the Naïve Bayes predictive model is a promising alternative approach. The Naïve Bayes Classifier did not reach a high accuracy, but other classifiers can be considered in the future. With GPR application on a larger scale across a range of diverse soil types, moisture levels, and cropping systems, larger data sets could also help calibrate models and raise the predictive accuracy. The model for predicting C structure could also benefit characterizing soil textures in the model. Extensive calibration and validation would thus improve mapping belowground C in more complicated conditions.

*Limitation of This Study*

As more experience is obtained from using GPR and better technology is developed, findings from studies similar to this investigation should provide better information pertaining to organic C in a variety of soil types. Scanning an empty sand trough before burying samples would further help identify sample locations. Additionally, having the samples buried 25.4 cm apart from each other complicated their positional separation in the data analysis. Lastly, the relatively shallow depth of sample placement in the trough affected the identification of the samples and also the sand surface on the B-scan images.

Moreover, the transmitters' and receivers' layout also significantly affected the results. Collaboration with scientists having an engineering background and experience would benefit future efforts.

Attribute analysis successfully correlated GPR data attributes with moisture levels. The next step could be to compare the performance of GPR data on different levels of soil moisture and soil–C mixtures to improve the correlation, as well as collecting 3-D GPR data to incorporate spatial analysis tools. Additionally, the attribute energy was proven to perform the best out of the four attributes. A replication of this research across years could further validate the consistency of the correlation coefficient. This would allow materials to be applied to more complicated soil dynamics to further assess the performance of the GPR data. Larger sample sizes could be achieved through implementing more channels as well as conducting multiple scans after adding moisture. After finding ideal attributes for correlation analyses and predictive modeling, scanning the trough with controlled C content could help accumulating training data points to build a baseline for each C content and increase accuracy. Towards scanning uncharacterized fields, GPR data could be able to correlate and predict C content. Extensive additional GPR screens across a range of diverse soil types would also facilitate in the development of robust, publicly available training data sets.

## 5. Conclusions

The Spearman's rank correlation tests across three moisture levels for all attributes were significant, indicating the ability of GPR to detect soil moisture content. As all of the coefficients were positive, this indicated that a higher moisture level was easier to be detected. The GPR was able to detect biochar-amended soil. With non-significant result from wavelet analysis, the Naïve Bayes predictive model identified C structures with 56% accuracy. The correlation coefficients between attributes and C content were still relatively small, indicating that quantification of C content is difficult to observe with varying water content using GPR. As for detecting moisture content and C content, the attribute energy performed the best among all attributes. This encourages future work to extract more informative feature like energy. Overall, these findings demonstrated the ability of GPR to detect minute objects such as organic C. This could extend future GPR applications into more complicated soil conditions and eventually towards characterizing both unknown soils and belowground biomass.

**Author Contributions:** Conceptualization, X.S., H.B., and R.J.; methodology, X.S., I.D.; software, X.S., I.D.; validation, X.S.; formal analysis, X.S.; writing—original draft preparation, X.S.; writing—review and editing, X.S., H.B., T.F., B.B., R.T.; visualization, X.S.; supervision, R.J.; project administration, X.S.; funding acquisition, D.H.

**Funding:** This research was funded by the Department of Energy of the United States (ARPA-E Award, No. DE-AR0000662).

**Acknowledgments:** We thank Isabel Morris and Branko Glišió and Andre Gonciar for providing inspiration for attributes, and the suggestion for computational calculating. We also thank Ilse Barrios Perez for providing Figure 3, Henry Ruiz for his contribution to the surface detection function, and Matthew Wolfe for assistance with the scanning cart operation.

**Conflicts of Interest:** The authors declare no conflict of interest. The funders had no role in the design of the study; in the collection, analyses, or interpretation of data; in the writing of the manuscript; or in the decision to publish the results.

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
