# Peer review of "Quantification of Soil Organic Carbon in Biochar-Amended Soil Using Ground Penetrating Radar (GPR)"

_remotesensing, doi:10.3390/rs11232874_

Round 1

Reviewer 1 Report

this is a well thought out research project.  I kept looking for some good images of profiles or "traces" or other data that can help the reader to understand amplitude, intensity and strength.  I am still a little confused about what those are, and exactly how you arrived at those values.  Also you have one sentence discussing the influence of water on this.  That seems to me to be the most important variable.   In all the analyses of carbon, have you accounted for differences in water?  And also, my understanding of GPR is that waves are produced from "differences" in the ground.  So what are those differences in your test plot?  Soil layers?  Moisture regimes?  That is also not clear to me.  But love your methods. 

Author Response

We appreciate and thank this reviewer for their time, effort, and helpful comments towards improving our manuscript. We have carefully read, discussed, and worked to incorporate all of the suggested revisions. For clarity, their comments (in blue text) and our responses (in black text) are below.

Reviewer 1’s comment

I kept looking for some good images of profiles or "traces" or other data that can help the reader to understand amplitude, intensity and strength.  I am still a little confused about what those are, and exactly how you arrived at those values. 

To better illustrate the methods, an illustrative figure of the experimental arrangement as well as a supplementary figure of a processed B-scan were added to the manuscript. An equation for the energy was also added to the manuscript.

Also you have one sentence discussing the influence of water on this.  That seems to me to be the most important variable.  

In this study, carbon content was the most important variable. The evaluation at 3 moisture levels was included and agreed in comparison with previous reports in which GPR could detect significant effects.

In all the analyses of carbon, have you accounted for differences in water? 

The analysis of carbon is the Pearson correlation, carbon content is the response and each attribute is the variable. GPR correlations with soil water content was done but separately.

And also, my understanding of GPR is that waves are produced from "differences" in the ground.  So what are those differences in your test plot?  Soil layers?  Moisture regimes?  That is also not clear to me. 

Uniform sand ‘parent’ soil in the experiment’s aboveground trough was used to minimize potential confounding variation from complex soil matrices. The differences detected by GPR in this study were between the samples’ carbon contents and carbon structures across treatments.

Reviewer 2 Report

The manuscript presents an application of Ground Penetrating Radar (GPR) for quantifying carbon content in soil. The quantification is explored through four attributes that are computed on A-scans, and analyzed using correlation and classification with Naïve Bayes. The results are not very high but seem promising. While the proposed application is interesting, there are several issues that should be addressed before the manuscript is ready for publication. I will deal with each section in more detail below, but most of the issues are related to missing or vague information.

SECTION 1: INTRODUCTION

The introduction is well written, but more focus should be put on the novelty of this work and the differences with existing works. Also, it would be interesting to add one of the GPR B-scans (or some A-scans) to show the difficulty of solving the problem using visual inspection.

SECTION 2: MATERIALS AND METHODS
Please include a table to summarize the 12 treatments, including at least the following fields for clarity: percentage of sand, percentage of biochar, percentage of graphite, percentage of activated carbon, and carbon content. This table would make Section 2.1 much clearer and help reference the different treatments throughout the text.

Similarly, please include a diagram of the trough in Section 2.2, including the buried treatments.

The attributes could be better defined, for instance, by adding equations or a more formal definition. An example of a great definition is Table 1 in

Morris, I., Abdel-Jaber, H., Glisic, B. (2019) Quantitative Attribute Analyses with Ground Penetrating Radar for Infrastructure Assessments and Structural Health Monitoring. Sensors 19(7), paper no. 1637.

Related to that, I am not sure I understand the definition of the energy and area. In the manuscript, energy is defined as the maximum integrated amplitude, whereas area is defined as the maximum integrated squared amplitude. However, these definitions are quite different from those in similar works, such as the one I referenced above and the following references from the manuscript:

Morris, I.; Glisic, B. GPR Attribute Analysis for Material Property Identification. Proceedings 2017 9th Int. Work. Adv. Gr. Penetrating Radar, IWAGPR 2017 – Proc. 2017, 1–5.

Dogan, M.; Turhan-Sayan, G. Preprocessing of A-Scan GPR Data Based on Energy Features. Detect. Sens. Mines, Explos. Objects, Obs. Targets XXI 2016, 9823 (May 2016), 98231E.

In the above references, energy is usually defined as the integrated squared amplitude, and area is usually defined as the integrated amplitude. As far as I know, this is the standard definition of both attributes; but it would seem to be the opposite definition to the one in this manuscript. Was there any confusion with the name of the attributes? Or was the definition based on a different reference?
Regardless of the definition, it is written that the attributes were transformations of "the central three A-scans of each sample." Does this mean that the four attributes were computed separately for each A-scan (12 attributes in total), or that the four attributes were computed somehow using all three A-scans?

There seems to be some confusion with the usage of the term "samples." In pages 1 through 5, samples is always used for the treatment samples, as far as I can gather. However, starting from page 6, the term seems to be used in its machine learning meaning (a data point to analyze). Please, separate these two meanings with two different words.

Related to that, I have been unable to understand how do the authors reach the total number of samples ("A total of 3868 samples were collected.", page 6, line 164). There are 12 treatments, 3 replications, and 7 channels... and 3868 is a multiple of none of these values. This should be clarified.

The Naïve Bayes classifier should be better explained, as well. The base assumption of Naïve Bayes is that the attributes are independent; however, the chosen attributes should be very much dependent (intensity is often equal to the square of the maximum amplitude, for one). Why not consider some other classifier that does not assume independence, such as classification trees or Linear Discriminant Analysis? Furthermore, such classifiers can consider the influence of multiple attributes jointly.

Some more specific comments or doubts:

Page 5, line 146: "Because of this, pre-processing procedures were reduced to a minimum including surface removal and fast-forward transfer bandpass filtering." Does this mean that the only pre-processing procedures were those two (surface removal and bandpass filtering), or were other procedures considered? Page 5, line 150: "This process removed the noise of the data above the sand surface and followed by sub setting the bottom of the B-scan by adding 50 rows from the sand surface." I am not sure I completely understand the second part of the procedure. What was the purpose for adding these rows, and why fifty? Also, were these rows the first 50 rows of the sand surface, or how were they chosen otherwise?

SECTION 3: RESULTS
The problem with the number of samples continues here, since it is unclear how do the authors reach 2208 samples after excluding 3/7 channels. Furthermore, the fact that removing channels results in a reduction of the number of samples implies that the attributes for different channels are considered as different samples. If this is the case, the authors should be careful with the splits for Naïve Bayes, since the same spatial position might have been analyzed by adjacent channels and thus the related A-scans (and attributes) will be more similar than expected. Assigning overlapping samples to both training and testing might artificially inflate classification performance.

With respect to the statistical analyses: moisture level (0%, 10%, 20%) is considered as a categorical variable, yet graphite carbon content (50%, 100%) is considered as a continuous variable. This is reflected by the usage of Spearman rank correlation (moisture level) versus Pearson correlation (graphite content). Could you explain this decision in text?

If I have understood it correctly, only the biochar treatments that were significantly different from 50% and 100% activated carbon were considered in the Pearson correlations shown in Table 3. However, this has probably inflated the correlations shown in Table 3, since the values have been selected to show differences. Could you show, or comment on, the results without this selection step?

As for the Naïve Bayes predictive model, all results are expressed in terms of area under the curve (AUC). I understand that these results have been obtained using the score for each one of the 12 classes. However, it would be interesting to know the performance of Naïve Bayes in terms of the accuracy (percentage of correctly classified samples) for the case with 12 classes. That would be a better indicator of performance in an actual setting, i.e., when we want to analyze the carbon content of an unknown sample. In that case, we would want only one result, the treatment, out of the 12 possible ones.

Also, I have been unable to find any mention of repetitions or iterations in the classification experiment. I assume this means the authors used only one repetition of the experiment, i.e., they split the data 75/25, trained and tested only once. Is this correct? If so, the results might not be stable or representative of the actual performance for unseen data. The authors should repeat the experiment multiple times (with a different 75/25 split each time) and average the results, to obtain a more representative result.

Some more specific comments or doubts:

Page 7, line 206: "When the attribute and moisture level associated monotonically, the coefficient became one." This sentence seems to imply that at least one correlation reached 1, but all the Tables show values lower than 0.5. Is this a problem with the tenses? Page 7, line 212: "Therefore, for all attributes, they increased when the moisture level increased, which indicates that GPR performed better at higher moisture levels." Technically, a positive correlation only means that the attributes increased when the moisture level increased. It is unclear what "performing better" means in this context. Page 7, line 230: "The area was estimated from a different strength of the signal amplitude." If I understood it correctly, the area was estimated as the (maximum) integrated square amplitude. However, this would seem similar to the energy (which is also an integral) and the intensity (which is also a function over the squared amplitude). Could you elaborate a bit more on this point? Page 7, line 233: The authors reference (Morris, 2017) -by the way, the cite is in the wrong format- to comment on the effect of the area, but that reference uses a different definition of area and energy than this work. In fact, their definition of area is closer to the definition of energy in this work. Thus, a direct comparison does not seem to be valid. Page 8, line 262: "The accuracy of predictive modeling ranged from 55% to 69% for each treatment (Table 4)." Table 4 shows AUC values, rather than accuracy. In machine learning, accuracy is usually defined as the percentage of correctly-classified samples.

SECTIONS 4 AND 5: DISCUSSION AND CONCLUSION 
In page 11, line 319, it is mentioned that "Additionally, having the samples buried 25.4 cm apart from each other complicated their positional separation in the data analysis." This is the first mention of such difficulty in text. Please, introduce this before the Discussion, possibly in Section 2.2 where the data collection procedure is explained.

Finally, the authors should probably moderate their comments on the performance of Naïve Bayes, since the obtained AUC values of 60% are still somewhat close to the trivial result (50%).

Some minor issues:

The authors should considering shortening the first half of the abstract, since it does not deal with their work and it takes the focus away from it. Page 4, line 124: There is no need to define acronym RCBD since the term only appears once. There are some references that do not conform to the format (e.g., (Delgado, 2019) in page 4, line 127). The authors might want to ensure Figure 1 is readable when printed in grayscale (the orange for the transmitters is quite similar to the background gray). Page 11, line 346: "The correlation coefficients between attributes and C content were mostly negative, indicating that quantification of C content is difficult to observe with varying water content using GPR." A negative correlation does not imply difficulty, but rather, an inverse relation rather than a direct relation. Low values (regardless of the sign) are indicators of difficulty, rather than negative values.

Author Response

We appreciate and thank the reviewer for their extensive time, effort, and helpful comments towards improving our manuscript. We have carefully read, discussed, and worked to incorporate all of the suggested revisions. For clarity, their comments (in blue text) and our responses (in black text) are below.

Reviewer 2 comments

The manuscript presents an application of Ground Penetrating Radar (GPR) for quantifying carbon content in soil. The quantification is explored through four attributes that are computed on A-scans, and analyzed using correlation and classification with Naïve Bayes. The results are not very high but seem promising. While the proposed application is interesting, there are several issues that should be addressed before the manuscript is ready for publication. I will deal with each section in more detail below, but most of the issues are related to missing or vague information.

SECTION 1: INTRODUCTION

The introduction is well written, but more focus should be put on the novelty of this work and the differences with existing works. Also, it would be interesting to add one of the GPR B-scans (or some A-scans) to show the difficulty of solving the problem using visual inspection.

Text further expressing the novelty of the work in regards to using GPR to estimate soil organic carbon (SOC) with biochar (no previous reports) has been expanded upon in the Introduction (lines 74-78).

A figure of processed B-scan and a heat map of B-scan (Supplementary Figure 1) has also been added to help visualize the data.

SECTION 2: MATERIALS AND METHODSPlease include a table to summarize the 12 treatments, including at least the following fields for clarity: percentage of sand, percentage of biochar, percentage of graphite, percentage of activated carbon, and carbon content. This table would make Section 2.1 much clearer and help reference the different treatments throughout the text.

A summary table has been added to the manuscript as Table 1.

Similarly, please include a diagram of the trough in Section 2.2, including the buried treatments.

The trough diagram has been added to the manuscript as Figure 1.

The attributes could be better defined, for instance, by adding equations or a more formal definition. An example of a great definition is Table 1 in

Morris, I., Abdel-Jaber, H., Glisic, B. (2019) Quantitative Attribute Analyses with Ground Penetrating Radar for Infrastructure Assessments and Structural Health Monitoring. Sensors 19(7), paper no. 1637.

An equation of the energy has been added to the manuscript.

Related to that, I am not sure I understand the definition of the energy and area. In the manuscript, energy is defined as the maximum integrated amplitude, whereas area is defined as the maximum integrated squared amplitude. However, these definitions are quite different from those in similar works, such as the one I referenced above and the following references from the manuscript:

Morris, I.; Glisic, B. GPR Attribute Analysis for Material Property Identification. Proceedings 2017 9th Int. Work. Adv. Gr. Penetrating Radar, IWAGPR 2017 – Proc. 2017, 1–5.

Dogan, M.; Turhan-Sayan, G. Preprocessing of A-Scan GPR Data Based on Energy Features. Detect. Sens. Mines, Explos. Objects, Obs. Targets XXI 2016, 9823 (May 2016), 98231E.

In the above references, energy is usually defined as the integrated squared amplitude, and area is usually defined as the integrated amplitude. As far as I know, this is the standard definition of both attributes; but it would seem to be the opposite definition to the one in this manuscript. Was there any confusion with the name of the attributes? Or was the definition based on a different reference?

Thanks for the clarification. Our definitions of energy and area were interchanged incorrectly. The two references were correct. Text in the results and conclusions sections regarding the two attributes has been changed accordingly.

Regardless of the definition, it is written that the attributes were transformations of "the central three A-scans of each sample." Does this mean that the four attributes were computed separately for each A-scan (12 attributes in total), or that the four attributes were computed somehow using all three A-scans?

The A-scans were selected for the middle three columns of each sample, which most accurately represent the most precise location of the samples in the trough. After computing the attributes on the three columns, their average provided the final attribute value. Precisely locating each sample was difficult using only the B-scan, so estimations for the sample locations were based on their burial positions. Since sample interval spacings were consistent, each sample bag was 19.05 centimeters (cm) in length, and the resolution of each pixel is 1 cm x 1cm, there would be approximately 19 A-scans present in each sample. To help ensure the utilized A-scans were located within samples, only the middle three columns of the sample were used to compute attributes.

There seems to be some confusion with the usage of the term "samples." In pages 1 through 5, samples is always used for the treatment samples, as far as I can gather. However, starting from page 6, the term seems to be used in its machine learning meaning (a data point to analyze). Please, separate these two meanings with two different words.

The ‘sample’ referred in the machine learning part was changed into ‘data point’.

Related to that, I have been unable to understand how do the authors reach the total number of samples ("A total of 3868 samples were collected.", page 6, line 164). There are 12 treatments, 3 replications, and 7 channels... and 3868 is a multiple of none of these values. This should be clarified.

The total number of the data points should be 4,536 calculated by 12 treatments, three replications, seven channels, three moisture levels, and six times scan per each moisture level. There were 668 data points missing because a few scans didn't make it to the end of trough, so some samples were not collected. This has added to the manuscript.

The Naïve Bayes classifier should be better explained, as well. The base assumption of Naïve Bayes is that the attributes are independent; however, the chosen attributes should be very much dependent (intensity is often equal to the square of the maximum amplitude, for one). Why not consider some other classifier that does not assume independence, such as classification trees or Linear Discriminant Analysis? Furthermore, such classifiers can consider the influence of multiple attributes jointly.

Thanks for the suggestion of the other two classifications. They are on our list of the machine learning approaches for future experiments. With our study being the first and only such report of GPR assays of carbon content, we believe the Naïve Bayes classifier is sufficient. However, we have changed the variable to predict the carbon from four attributes to only the energy and repeated the analysis to help address this concern. Moreover, wavelet analysis was attempted and added to the manuscript to assess its utility as an alternative.

Some more specific comments or doubts:

Page 5, line 146: "Because of this, pre-processing procedures were reduced to a minimum including surface removal and fast-forward transfer bandpass filtering." Does this mean that the only pre-processing procedures were those two (surface removal and bandpass filtering), or were other procedures considered?

For this research, only surface removal and fast-forward bandpass filtering were applied. Other pre-processing procedures (gain correction, migration, and Hilbert) were attempted but didn’t improve the results or B-scan images. As a result, we decided to keep the pre-processing procedures to a minimum.

Page 5, line 150: "This process removed the noise of the data above the sand surface and followed by sub setting the bottom of the B-scan by adding 50 rows from the sand surface." I am not sure I completely understand the second part of the procedure. What was the purpose for adding these rows, and why fifty? Also, were these rows the first 50 rows of the sand surface, or how were they chosen otherwise?

Yes, the rows are the first 50 rows below the sand surface to ensure inclusion of the region where sample bags were buried. Sample burial depth was consistent but subject to minute variation.

SECTION 3: RESULTSThe problem with the number of samples continues here, since it is unclear how do the authors reach 2208 samples after excluding 3/7 channels.

In total, channel 1 collected 558 data points, channel 6 collected 558 data points, and channel 7 collected 544 data points. Subtracting them from 3,868 produced the final count of 2,208.

Furthermore, the fact that removing channels results in a reduction of the number of samples implies that the attributes for different channels are considered as different samples. If this is the case, the authors should be careful with the splits for Naïve Bayes, since the same spatial position might have been analyzed by adjacent channels and thus the related A-scans (and attributes) will be more similar than expected. Assigning overlapping samples to both training and testing might artificially inflate classification performance.

The attributes for different channels are not different, as the antenna was designed to maximize array density and potential for improved spatial resolution. Channel removal was based on extreme outliers (IQR) on the outermost edge of the aboveground trough and most likely caused by lack of sufficient parent soil (sand) width to ensure even signal return. With such a dense, multichannel array, however, the potentially overlapping data for Naïve Bayes was considered. As a result, reanalysis was performed with the middle channel, channel 4’s data points only utilized to train and test the model.

With respect to the statistical analyses: moisture level (0%, 10%, 20%) is considered as a categorical variable, yet graphite carbon content (50%, 100%) is considered as a continuous variable. This is reflected by the usage of Spearman rank correlation (moisture level) versus Pearson correlation (graphite content). Could you explain this decision in text?

One of the goals of this experiment was to determine continuous correlation between the carbon and GPR data. For biochar, we included 2%, 4%, 6%, 8%, 10%, 50%, and 100% treatment levels to create continuous variable. Activated carbon carbon content and graphite carbon content were categorical. This resulted in a mixed model, but this research was mainly focused on biochar with activated carbon and graphite used to compare the carbon structure. With this trade off in consideration, we selected the Pearson correlation to describe the relationship between carbon and attributes.

If I have understood it correctly, only the biochar treatments that were significantly different from 50% and 100% activated carbon were considered in the Pearson correlations shown in Table 3. However, this has probably inflated the correlations shown in Table 3, since the values have been selected to show differences. Could you show, or comment on, the results without this selection step?

The Pearson correlation for all activated carbon and biochar together was performed but found to be non-significant. The rationale for selecting the biochar was not to use this model for future research but rather to demonstrate the possibility of correlating carbon content with GPR data.

As for the Naïve Bayes predictive model, all results are expressed in terms of area under the curve (AUC). I understand that these results have been obtained using the score for each one of the 12 classes. However, it would be interesting to know the performance of Naïve Bayes in terms of the accuracy (percentage of correctly classified samples) for the case with 12 classes. That would be a better indicator of performance in an actual setting, i.e., when we want to analyze the carbon content of an unknown sample. In that case, we would want only one result, the treatment, out of the 12 possible ones.

Thanks for pointing out the difference between AUC and accuracy. We have corrected the term ‘accuracy’ throughout.

Also, I have been unable to find any mention of repetitions or iterations in the classification experiment. I assume this means the authors used only one repetition of the experiment, i.e., they split the data 75/25, trained and tested only once. Is this correct? If so, the results might not be stable or representative of the actual performance for unseen data. The authors should repeat the experiment multiple times (with a different 75/25 split each time) and average the results, to obtain a more representative result.

The initial iterations were less than 10, and the 75/25 split was different each time. To improve the representative stability of this work, the iterations were increased to 1000 and re-analyzed. The methods and results sections for the Naïve Bayes have been changed accordingly.

Some more specific comments or doubts:

Page 7, line 206: "When the attribute and moisture level associated monotonically, the coefficient became one." This sentence seems to imply that at least one correlation reached 1, but all the Tables show values lower than 0.5. Is this a problem with the tenses?

This sentence meant to illustrate the Spearman rank correlation potential relationships, but our did not have a correlation coefficient of 1.

Page 7, line 212: "Therefore, for all attributes, they increased when the moisture level increased, which indicates that GPR performed better at higher moisture levels." Technically, a positive correlation only means that the attributes increased when the moisture level increased. It is unclear what "performing better" means in this context.

This sentence was clarified from ‘performing better’ to ‘data had a positive relationship with moisture level’.

Page 7, line 230: "The area was estimated from a different strength of the signal amplitude." If I understood it correctly, the area was estimated as the (maximum) integrated square amplitude. However, this would seem similar to the energy (which is also an integral) and the intensity (which is also a function over the squared amplitude). Could you elaborate a bit more on this point?

The attribute area was computed by integrated the max amplitude, and the result for the attribute area was changed accordingly. As the statement before, it’s a definition mistake made by the authors and has been corrected in the text.

Page 7, line 233: The authors reference (Morris, 2017) -by the way, the cite is in the wrong format- to comment on the effect of the area, but that reference uses a different definition of area and energy than this work. In fact, their definition of area is closer to the definition of energy in this work. Thus, a direct comparison does not seem to be valid.

The comparison was removed after reanalysis of the data.

Page 8, line 262: "The accuracy of predictive modeling ranged from 55% to 69% for each treatment (Table 4)." Table 4 shows AUC values, rather than accuracy. In machine learning, accuracy is usually defined as the percentage of correctly-classified samples.

The usage of ‘accuracy’ and ‘AUC value’ have been corrected.

SECTIONS 4 AND 5: DISCUSSION AND CONCLUSION In page 11, line 319, it is mentioned that "Additionally, having the samples buried 25.4 cm apart from each other complicated their positional separation in the data analysis." This is the first mention of such difficulty in text. Please, introduce this before the Discussion, possibly in Section 2.2 where the data collection procedure is explained.

The difficulties were added and explained in section 2.2.

Finally, the authors should probably moderate their comments on the performance of Naïve Bayes, since the obtained AUC values of 60% are still somewhat close to the trivial result (50%). 

The result and discussion section for Naïve Bayes model has changed, and the interpretation term has changed to accuracy obtained from the confusion matrix.

Some minor issues:

The authors should considering shortening the first half of the abstract, since it does not deal with their work and it takes the focus away from it.

For many agronomists who focus on SOC, they benefit from more general GPR information than people who have worked with GPR. Our abstract attempts to balance the importance of SOC and how GPR may help with it to a broad audience. We still agree that this comment has value, so one sentence emphasizing on the commercial potential of biochar has removed.

Page 4, line 124: There is no need to define acronym RCBD since the term only appears once.

The RCBD acronym has been removed.

There are some references that do not conform to the format (e.g., (Delgado, 2019) in page 4, line 127).

The reference format has been corrected.

The authors might want to ensure Figure 1 is readable when printed in grayscale (the orange for the transmitters is quite similar to the background gray).

The layout of the Figure 1 (now Figure 2) has been improved.

Page 11, line 346: "The correlation coefficients between attributes and C content were mostly negative, indicating that quantification of C content is difficult to observe with varying water content using GPR." A negative correlation does not imply difficulty, but rather, an inverse relation rather than a direct relation. Low values (regardless of the sign) are indicators of difficulty, rather than negative values.

This line changed the justification of ‘negative value’ to ‘relatively small value’.

Round 2

Reviewer 2 Report

The changes made by the authors have much improved the manuscript, making it clearer and easier to understand. I believe the manuscript is basically ready for publication, but I will mention some very small issues that should be solved beforehand.

The response of the authors has been quite informative and it has helped me to better understand the manuscript. However, there are several clarifications in the answers that I have been unable to find in text. For instance:

The authors clarified that when they say they transformed the central three A-scans, they mean they extracted the attributes separately for each A-scan and then averaged them across the three A-scans. However, I have been unable to find any mention to this averaging in text. The reasons for adding 50 rows at the bottom of the B-scan was clarified, but I have been unable to find this explanation in text.

These comments should be added to the text, since they help clarify and motivate several of the decisions made by the authors.

Also, very minor typo: the font in lines 135-138 is different from the one in the rest of the manuscript.

Author Response

For clarity, reviewer’s comments are in blue text and our responses (in black text) are below.

The changes made by the authors have much improved the manuscript, making it clearer and easier to understand. I believe the manuscript is basically ready for publication, but I will mention some very small issues that should be solved beforehand.

We would like to gratefully thank the reviewer for reviewing our revisions in such a short time, and we’re also thankful for the great help that the reviewer has provided to improve the manuscript.

The response of the authors has been quite informative and it has helped me to better understand the manuscript. However, there are several clarifications in the answers that I have been unable to find in text. For instance:

The authors clarified that when they say they transformed the central three A-scans, they mean they extracted the attributes separately for each A-scan and then averaged them across the three A-scans. However, I have been unable to find any mention to this averaging in text.

Additional text to help explain the justification for A-scan calculations has been added in lines 157-160.

The reasons for adding 50 rows at the bottom of the B-scan was clarified, but I have been unable to find this explanation in text.

Justification for the inclusion of 50 rows has been added in lines 193-195.

These comments should be added to the text, since they help clarify and motivate several of the decisions made by the authors.

Also, very minor typo: the font in lines 135-138 is different from the one in the rest of the manuscript.

The font in lines 135-138 and the entire manuscript has been checked and corrected.
